# Status of Phytotoxins Isolated from Necrotrophic Fungi Causing Diseases on Grain Legumes

**DOI:** 10.3390/ijms24065116

**Published:** 2023-03-07

**Authors:** Francisco J. Agudo-Jurado, Pierluigi Reveglia, Diego Rubiales, Antonio Evidente, Eleonora Barilli

**Affiliations:** 1Plant Breeding Department, Institute for Sustainable Agriculture (CSIC), 14004 Córdoba, Spain; 2Department of Chemical Sciences, University of Naples Federico II (UNINA), 80138 Naples, Italy; 3Institute of Sciences of Food Production, National Research Council, 70126 Bari, Italy

**Keywords:** necrotrophic fungi, fungal phytotoxins, secondary metabolites, legumes

## Abstract

Fungal phytotoxins can be defined as secondary metabolites toxic to host plants and are believed to be involved in the symptoms developed of a number of plant diseases by targeting host cellular machineries or interfering with host immune responses. As any crop, legumes can be affected by a number of fungal diseases, causing severe yield losses worldwide. In this review, we report and discuss the isolation, chemical, and biological characterization of fungal phytotoxins produced by the most important necrotrophic fungi involved in legume diseases. Their possible role in plant–pathogen interaction and structure–toxicity relationship studies have also been reported and discussed. Moreover, multidisciplinary studies on other prominent biological activity conducted on reviewed phytotoxins are described. Finally, we explore the challenges in the identification of new fungal metabolites and their possible applications in future experiments.

## 1. Introduction

Legumes are members of the third largest plant family, Fabaceae, with over 20,000 species, many of which were domesticated at the very onset of agriculture. Among them, grain legumes are important food and feed crops, having played a key role as the basis of the food of all major civilizations combined with cereals [1], being staple crops in many regions [2], and also have a major role in animal feeding [3]. Legumes are also gaining in popularity due to their various health [4,5] and environmental [6] benefits, as their ability to fix atmospheric nitrogen through a symbiotic relationship with *Rhizobium*, which can be used by the crop itself or left in the soil for subsequent crops, makes legumes essential for sustainable agriculture. 

The demands for protein crops are markedly increasing [7]; this should be paired with technical solutions to support cultivation. As with any crop, legumes can be affected by biotic or abiotic stresses, reducing their quality and yield, which translates into economic losses for the farmer. Biotic stresses can be caused by a range of biological organisms such as bacteria, viruses, fungi, insects, nematodes, or even parasitic plants [8]. Among these, fungal diseases are likely to be the most economically relevant threats. Indeed, they have been a devastating menace throughout history, with vast epidemics and disastrous yield losses still occurring [9]. 

Regardless of their lifestyle, all fungi can be recognized by plant immune systems and elicit several host defense responses. The plant’s innate immune system can display two layers of defenses: pathogen-associated molecular pattern (PAMP)-triggered immunity (PTI), and effector-triggered immunity (ETI). PTI is the first line of defense reactions. It is initiated in plants when PAMPs are recognized by membrane-localized pattern recognition receptors (PRRs) [10]. When PAMPs are recognized through PRRs, they trigger a fast, relatively weak but broad-spectrum immune response to pathogen infection. This response includes the accumulation of reactive oxygen intermediates, the accumulation of antimicrobial compounds, changes in hormone biosynthesis such as salicylic acid, jasmonates, or ethylene, and plant cell wall reinforcement [11,12,13]. PTI is effective against non-adapted fungi [14]. 

ETI is the second line of innate immunity. Effectors that trigger ETI are usually highly specific and perceived by plant resistance proteins (*R* proteins). In this case, the recognition leads to a rapid, robust, and localized cell death response, often referred to as a hypersensitive reaction (HR) [12].

Conversely, the objective of a fungus is to obtain nutrients from the host plant by penetrating and neutralizing plant defenses [15]. According to the colonization/infection strategy, fungal pathogens can be generally classified as biotrophic or necrotrophic [16]. Biotrophic fungi feed on living cells, so the penetration structures allow the invading fungus to suppress the plant’s immune response system and for the reprogramming of its metabolism [17]. Alternatively, necrotrophic fungi emit a number of compounds to degrade the host tissues and obtain nutrients [18]. In order to establish a compatible interaction leading to its proliferation, the fungus must avoid eliciting PTI or either cope with or suppress it. For this purpose, necrotrophic fungi can inactivate the plant defenses by secreting toxic compounds. These compounds are mainly enzymes, catalyzing the degradation of structural components and other essential substances, and phytotoxins, causing damage and alterations in the cells.

Phytotoxins can be either host specific toxins (HSTs) that affect only a particular plant species or more often genotypes of that species [19], or non-host specific toxins (nHSTs) that affect a broad range of plant species. Induction of host alterations such as DNA damage, abnormal mitochondrial oxidation, cytotoxicity, etc. with the goal of leading to host cell death are some of the common functions of HSTs [20]. These HST toxins are diverse, chemically ranging from low-molecular-weight compounds to cyclic peptides [21,22,23,24]. High molecular weight compounds such as some polysaccharides have also been reported as phytotoxins, although their true role is an open question [25,26,27]. Genes encoding polypeptides for the biosynthesis of these HSTs have been shown to reside on a conditionally dispensable chromosome that controls host-specific pathogenicity [28]. The mechanism of host-selective pathogenesis, through the HSTs, is well-understood and documented [29,30]. In some cases, host sensitivity was mediated by gene-for-gene interactions, and the toxin sensitivity was mandatory for disease development. 

Phytotoxins belong to different chemical families and are commonly classified as polyketides, macrolides, non-protein amino acids, naphthalenones, anthraquinones, furanones, pyranones, non-ribosomal peptides, alkaloids, terpenes, or metabolites of mixed biosynthetic origin [21,22,23,24,30,31,32]. This structural diversity is related to the complexity of the host–pathogen relationship. Assigning the correct structure to a fungal phytotoxin is a stepping stone to shed light on its biosynthesis pathways and their regulation, the mode of action, and how this relates to fungal virulence [33,34]. 

Recent advances in organic and analytical chemistry techniques such as nuclear magnetic resonance (NMR) and liquid chromatography coupled with mass high-resolution mass spectrometry (LC-MS), together with the development of software and bioinformatic tools, have allowed for the isolation and structural elucidation of low abundant and novel phytotoxins [21,22,23,24,35,36,37,38].

With a particular focus on necrotrophic fungi, important grain legumes such as chickpea (*Cicer arietinum*), faba bean (*Vicia faba*), field pea (*Pisum sativum*), lentil (*Lens culinaris*), lupin (*Lupinus* spp.), common bean (*Phaseolus vulgaris*), or soybean (*Glycine max*) are impacted by pathogens belonging to the *Ascochyta* spp., *Botrytis* spp., *Colletotrichum* spp., *Phoma* spp. or *Macrophomina* species. They cause significant economic losses [39] through direct seed loss and reduced marketability as well as the costs derived from disease management including chemical and cultural methods. Disease resistance is currently a primary objective of most plant breeding programs. Durable and multi-disease resistance is considered as a prerequisite to broad environmental adaptation aiming at stabilizing agricultural systems.

Herein, we review the isolation, chemical, and biological characterization of fungal phytotoxins produced by the most important necrotrophic fungi involved in legume diseases. In addition, research on the mode of action of the reviewed phytotoxins will also be discussed. Finally, multidisciplinary studies on other prominent biological activity conducted on the reviewed phytotoxins are also reported.

## 2. Phytotoxic Metabolites Isolated from Necrotrophic Pathogenic Fungi on Grain Legumes Crops

The most important pathogenic necrotrophic fungi of grain legumes are *Ascochyta*, *Colletotrichum* or *Botrytis*, together with their anamorphs [40], followed by *Cercospora*, *Macrophomina*, *Pleiochaeta*, or *Sclerotinia*, as listed in Table 1. 

### 2.1. Phytotoxic Compounds Produced by Ascochyta *spp.*

*Ascochyta* species cause diseases globally called Ascochyta blights, whose symptoms typically develop in the aerial parts of the plants under high humidity and average temperature conditions, producing necrotic lesions on the leaves and stems [66]. Leaves with many lesions wither before the lesions become large, especially on the lower portion of the plants. On the stems, these fungi cause deep necrotic lesions that can lead to the breaking of stems and the death of plant parts above the affected zone. Infected grains and pods can spread disease through seeds, causing their use in the following crops to be harmful since they can drown growing plants. Ascochyta blights are incited by different pathogens in the various legumes such as *Ascochyta lentis* in lentil; *A. pisi* and *A. pinodes* in pea; *A. lentis* var. *lathyri* in grass pea; *A. fabae* in faba bean; and *A. rabiei* in chickpea [67]. Ascochyta blight remains an extremely difficult pathogen to control, primarily due to the limited levels of host resistance available, and secondarily, because fungicides are often uneconomic [68], forcing the integration of the use of genetic resistance with cultural practices. Therefore, the main disease control strategy has been to avoid sowing close to infested field stubbles and/or to delay the sowing of field crops for as long as possible. This minimizes inoculum carry-over and its survival on crop residues and in soil, avoiding the initial infection of the crop from aerial inoculum arising from infested residues [69,70,71]. Nevertheless, late sowing is not an option in some countries due to the short crop season, and this practice incurs unsustainable yield penalties in many instances. 

Different metabolites with pathogenesis-determining cytotoxic capacity have been reported in several *Ascochyta* species, as listed in Table 1. For instance, 10 metabolites have been isolated and identified from *A. lentis*, being lentiquinones A–C, lentisone, ω-hydroxypachybasin, 1,7-dihydroxy-3-methylanthracene-9,10-dione, phomarin, pachybasin, tyrosol, and pseurotin A (1–10, Figure 1). However, between them, only compounds 1–4, 8, and 9 showed phytotoxic activity on the lentil plants, being also capable of reducing the root growth and seed germination; in contrast, the reported activity of metabolites 5–7 was almost null in the bioassay condition [41,72], and the role they play in plant–pathogen interaction is not well-defined yet [73,74].

The so-called Ascochyta blight of peas is in fact a disease complex that can be caused by several fungi including *A. pisi*, *A. pinodes*, and *Phoma medicaginis*. Compounds described so far in *A. pinodes* with noteworthy phytotoxic activity have been pinolidoxin, 7-*epi*-pinolidoxin, 5,6-dihydropinolidoxin, 5,6-epoxypinolidoxin, herbarumin II, 2-*epi*-herbarumin II, and pinolide (**11–17**, Figure 2) [42,43,44]. Between them, pinolidoxin (**11**) showed the highest phytotoxic activity measured in pea plants such as lesion size (mm^2^) on both the pods and in leaves as well as in other grain legumes such as faba bean. In contrast, the other metabolites found only produced reduced symptoms [42,44]. When the compounds were tested on other legumes, it was observed that herbarumin II, 2-*epi*-herbarumin II, and pinolide (**15** and **17**) did not display significant phytotoxic activity. The importance of the stereochemistry of the hydroxy group at C-7 of these compounds on phytotoxicity was deduced by the authors [44].

Ascosalitoxin (**18**, Figure 3), which is a derivative of salycilic aldehyde, was isolated as the main phytotoxin from *A. pisi* [75]. Ascosalitoxin (**18**) displayed phytotoxic activity on pea and faba bean leaves and pods, and on tomato seedlings [76]. Ascochitine, an *o*-quinone methide, is an abundantly produced phytotoxin that was first discovered in culture extracts of *A. pisi* [77], and later in *A. fabae* [78] (**19**, Figure 3), where it displayed antibiotic activity. 

More recently, ascochitine (**19**, Figure 3) was found in the culture extracts of many wild vetch-infecting *Ascochyta* and *Ascochyta*-like species including *A. viciae-villosae* [78]. The widespread distribution of ascochitine (**19**) production indicates its ancient origin in these related taxa. Ascochitine (**19**) production is not restricted to the legume-associated *Ascochyta* species, but also to some *Phoma* species. In phytotoxicity studies performed on faba bean plants, ascochitine (**19**) was shown to produce electrolyte leakage when tested on leaf discs as well as necrosis and wilting in whole plants assays [45].

Concerning *A. lentis* var. *lathyri*, the compounds described have been lathyroxins A and B, *p*-hydroxybenzaldehyde, *p*-methoxyphenol (**20–23**, Figure 4), and tyrosol (**9**, Figure 1) [22]. The latter has also been isolated in *A. pinodes* [41]. Lathyroxin B (**21**) showed activity in a panel of legumes tested including lupine, lentil, and beans. In contrast, lathyroxin A (**20**) only showed activity on lupin and *Sonchus oleraceus*, while *p*-hydroxybenzaldehyde (**22**) was toxic only on lupin and lentil [22].

Finally, compounds described for *Ascochyta rabiei* were solanapyrone A [47], solanapyrone B [48], solanapyrone C [47], and cytochalasin D [49,79] (**24–27**, Figure 5). The solanapyrones A–C (**24–26**) are structural isomers that act as HSTs [44]. When tested on whole plant assays in chickpea plants, solanapyrones A, B, and C were shown to be active individually as well as in combination, being able to reduce root development as well as the seed germination [80] of the host plant.

#### Structure–toxicity relationship studies of phytotoxins produced by *Ascochyta* spp.

Although a significant number of compounds produced by *Ascochyta* spp. have been identified, only a few toxicity relationship studies have been carried out with them. For example, the nonenolide pinolidoxin (**11**) is the main phytotoxin produced by both *Ascochyta* spp., which is closely related to putaminoxin, having a similar nonenolide ring system and some substituent groups. *Phoma putaminum* is a fungus proposed for the biocontrol of the dangerous weed *Erigeron annus*. The two nonenolides, pinolodoxin and putaminoxin, along with some of their natural analogues and some hemisynthetic derivatives, were assayed for their phytotoxic, antifungal, and zootoxic activities. The results obtained by testing all of the compounds on the weeds and crops showed that the phytotoxic activity was related to the integrity of the nonenolide ring, to the presence of two hydroxyl groups, and to an unmodified propyl side chain. Likewise, pinolidoxin (**11**) was detected in *A. pinodes* in front of the hyphae as it developed, which suggests that this compound has a fundamental role in modulating the defense response in plants [81]. Furthermore, among a set of phytotoxins with different carbon skeletons and produced by different pathogenic fungi, only the nonenolides pinolidoxin and putaminoxin appeared to be inhibitors of the first steps in the phenylpropanoid pathway. This is the route in charge of generating compounds such as phytoalexins or lignin in the defense against parasitic attacks [82]. 

Structure–activity relationship studies have been also performed with compounds belonging to the Solanapyrone group (**24–26**). In fact, solanapyrone A (**24**) was shown to be active against *Bacillus subtilis* and *Micrococcus tetragenus*, in addition to certain saprobe fungi [83], while solanapyrone C (**26**) only acted against *B. megaterium*, and a unicellular alga [83,84]. Solanapyrone A (**24**) binds specifically to DNA polymerases [85], acting in cell control during mitosis and meiosis, postulating that this compound could inhibit DNA repair processes, unbalancing the cell cycle, and finally causing apoptosis; or by affecting defense signaling induced by DNA damage and the subsequent repair process [86,87]. Solanapyrones J and K had activity against Gram-positive bacteria, but not against Gram-negative bacteria. However, when solanapyrones L and M, which differ in the functionalization of the pyrone ring, were tested, the latter did not show any activity [88].

### 2.2. Phytotoxic Compounds Produced by Botrytis *spp.*

Chocolate spot can be elicited by both pathogens, *Botrytis fabae* and *B. cinerea*, but *B. fabae* is more harmful to faba bean [89]. Chocolate spot is an important disease, having a worldwide distribution and causing a series of dark brown spots on the aerial parts of plants [90]. When the humidity reaches high levels and there is an average temperature of around 22 °C, the fungus begins an aggressive phase where it spreads very quickly, significantly increasing the number of necrotic spots and withering the plant completely in a period of two days in some cases [40]. When the plant is affected during the flowering period, the flowers fall, decreasing the final yield production and favoring the spread of the fungus to the lower parts of the host or to other neighboring plants. In addition, if the pathogen affects the pod during its formation, the seed quality decreases, and often, their commercialization is unviable [91].

The pathogen should be limited by applying both agronomic control including removing the crop infected remains and their destruction as well as by chemical control techniques through the utilization of fungicides with different chemical characteristics such as benzimidazoles (benomyl, carbendazim) or dithiocarbamates (mancozeb) among others [70,92]. However, these methods are usually very costly, which is why other strategies have been formulated such as the use of resistant varieties obtained through the development of plant breeding programs. Although in the last 20 years some resistant materials have been described [93], more effort is needed in order to incorporate resistance into commercial varieties as well as testing the stability of sources of resistance through time and space [88].

Reported compounds produced by *B. fabae* are botrytone, regiolone, *cis*-2,4,8- and *trans*-2,4,8-trihydro-1-tetralone, (4*S*)-(+)-isosclerone, scytalone, and 3-hydroxyjuglone (**28–34**, Figure 6). Out of these metabolites, botrytone (**28**) has shown some phytotoxicity on the host plant, with regiolone, *cis*-2,4,8, and *trans*-2,4,8 trihydro-1-tetralone (**29–31**) being the most toxic [50].

Because of the diverse host range that *B. cinerea* presents, most of the phytotoxins that have been identified from this fungus have been extracted from other hosts including legumes such as the common bean (*P. vulgaris*), or other different species such as sweet pepper (*Capsicum annuum*), although later tests have been extrapolated to legumes. Compounds identified in *B. cinerea* (Table 1) include botrydial, botryendial, botrydienal, 8,9-*epi*-botrydial, 1-*epi*-botrydial, dihydrobotrydial, norbotrydialone acetate; 10-oxodihydrobotry-1(9),4(5)-diendial, 10-oxodehydrodihydrobotrydial, 4β-acetoxytetrahydrobotryslactone, botryenalol, β-O-methyldihydrobotridialone, botrylactone, botcinic acid, 3-acetylbotcinic acid, and botcinin A (**35–50**, Figure 7). Out of these compounds, botrytone, regiolone, *cis*- and *trans*-2,4,8-trihydroxy-1-tetralone botrydial, botryendial, botrydienal, 8,9-*epi*-botrydial, and 4β-acetoxytetrahydrobotryslactone (**28–31**; **35–38**, and **44**) showed phytotoxic activity when tested on the host both in the cut leaf and in the whole plant. Bioassays using *B. cinerea* mutants deficient in the production of botrydial and botcinin A (**35** and **50**) showed no reduction in pathogenicity, being capable of damaging cells of the host plant tissue. Moreover, a marked lower virulence of the fungus was demonstrated [54]. Phytotoxic activity of some metabolites such as botrydial (**35**) has been shown to be influenced by external factors such as light intensity [52].

#### Structure–toxicity relationship studies of Phytotoxins from *Botrytis*
*spp.*

Botrydial (**35**) was the only phytotoxin produced by *Botrytis cinerea* for which structure–activity relationship studies were conducted. This compound induces a hypersensitive response in the host, which is regulated via the salicylic acid and jasmonic acid pathways [94]. It has been shown that the activity of this compound and its epimers is closely related to the C-1 and C-8 carbons, depending on the oxidation states of the aldehyde substituents as well as the C-9 carbon, observing a lower activity, for example, in botryendial and botrydienal (**36** and **37**) with respect to botrydial (**35**). Likewise, the configuration (*S*) at the C-1 carbon has been observed to be critical in the substrate–receptor role [95].

### 2.3. Phytotoxic Compounds Produced by Macrophomina *spp.*

Several species of *Macrophomina* cause a disease called “charcoal rot” on different hosts [96,97]. Here, we will only deal with *M. phaseolina* causing charcoal rot on soybean, although it can also affect other legume crops such as cowpea, common bean [98], and other plant species as strawberry [99] or sunflower [100]. The pathogen is widely distributed in different parts of the world, indicating its omnipresence in varied soil types. Disease symptoms are more severe under dry and warm (28–35 °C) growing conditions, with soil being the principal source of inoculum. Infection is started by the spores that survive in the soil or in the remains of infected plants and develops in response to root exudates of the host plants. Infection causes an abnormal development of the plant and chlorosis in the leaves, ending in the total wilting of the plants [96]. 

The management of this disease is quite complex, since an integrated approach is necessary to reduce the number of viable spores in the soil or in the material that is used, sowing clean seeds or adopting crop rotations with resistant material, because fungicides are not fully effective against the pathogen [101,102]. A recent meta-analysis on biological control methods highlighted that *Trichoderma gamsii*, *Gliocladium virens*, *Trichoderma viride*, and *Pseudomonas fluorescence* have a higher control efficiency [103]. Nevertheless, the search for genetic resistance in the crop is still scarce [104].

Phytotoxic metabolites produced by *M. phaseolina* have been described including phaseolinone, (-)-botryodiplodin, phaseocyclopentenones A and B, and guignardone A [55,56,57] (**51–55**, Figure 8). Phaseolinone (**51**) and (-)-botryodiplodin (**52**) are believed to play a role in the initial stages of infection, causing the wilting of seedlings and the formation of necrotic lesions on the leaves and roots [55]. This increases the virulence of *M. phaseolina* and may help to explain the highly efficient mechanism to infect different hosts and tissues. Likewise, although it has not yet been specified in a concrete way, it is speculated that the variation in the production of (**51**) and (**52**) between different isolates may be due to the geographical variation in the isolates due to different environmental conditions or the production and interaction with other phytotoxins [56].

Phaseocyclopentenones A and B (**53** and **54**) were recently described together with guignardone A (**56**), extracted from a strain of *M. phaseolina* isolated in infected soybean tissues from Argentina. Compounds **53–55** showed phytotoxic activity assayed on tomato plants, used as a non-host control by the leaf puncture assay, while only **53** and **54** were toxic when tested on cuttings of the same plant. No antifungal activity was detected for the three metabolites against some fungal pathogens such as *Cercospora nicotianae* and *Colletotrichum truncatum*, which are two severe pathogens both isolated from infected soybean plants in Argentina [57]. 

#### Structure–toxicity relationship studies of phytotoxins from *Macrophomina* spp.

Phaseolinone (**51**) is considered a mutagenic compound due to its primary and secondary alcoholic groups. When these hydroxyl groups are modified, a reduction in mutagenic activity is observed when one of them is replaced (with a ketone group), and the complete loss of this activity when a complete substitution of the hydroxyl groups is performed. In the same way, a reduction in the toxicity of the molecule was also observed as more hydroxyl groups are substituted, suggesting that side-chain epoxide and alcoholic groups are essential for its activity [105]. Additionally, (-)-botryodiplodin (**52**), which is a natural analogue of ribose, interferes with different cellular mechanisms such as transporters or enzymes, although it is not entirely clear which. An example of this interference may be the absence of a hydroxyl group at C-5, causing it not to be an optimal substrate for the ribose 5-kinase enzyme, or it may exert another function when present in the cell cytoplasm in its phosphorylated form [106].

Phytotoxicity assays have shown that the functionalization of C-4 and C-5 in phaseocyclopentenones A (**53**) and B (**54**) are important for phytotoxicity. In addition, these compounds showed a different mechanism of action depending on the bioassay condition [57]. The complete phytotoxicity assays of guignardone A (**55**) in legumes have not been carried out yet.

### 2.4. Phytotoxic Compounds Produced by Colletotrichum *spp.*

Species of the anamorphic genus *Colletotrichum* (teleomorph *Glomerella*) are implicated in plant diseases, generally referred to as anthracnoses, which are found throughout the world. The various *Colletotrichum* species include some of the most destructive post-harvest pathogens that can affect a multitude of hosts including cereals, legumes, fruits, and vegetables [107]. *Colletotrichum* spp. can survive for several years on plant debris that remains in the field after harvest [108]. The pathogen requires more than 16 h of leaf wetness in combination with temperatures between 20 and 30 °C to infect the host plant [107]. Initial symptoms on leaves are small yellow spots that enlarge into brown-colored lesions with a distinct dark margin. This might result in premature leaf drop. In the stem, the first lesions appear in its base from where they progress upwards [109]. Large stem lesions can surround the whole stems and penetrate the vascular tissue, causing wilting with subsequent plant death. In susceptible genotypes, more than 20% of the harvested seeds could show necrotic lesions, affecting their quality and market sale. During the growing season, the inoculum is primarily spread by rain splash and secondarily by windblown infected debris or during the harvesting process [109]. Chemical control through the application of fungicides is normally used, but this can generate some resistance in the fungus and can lose effectiveness over time. Biological control is also an alternative method, which uses various antagonistic microorganisms such as *Bacillus subtilis* [110]. In legumes, genetic sources of resistance have been developed in common bean, soybean, and lentil [111,112,113], where some varieties with partial resistance to the disease have been identified. 

*C. truncatum* is the main causal agent of soybean anthracnose, which is characterized by pre- and postemergence damage on cotyledons, pods, petioles, and stems. The metabolites produced by *C. truncatum* are colletruncoic acid methyl ester and *meso*-2,3-butane-2,3-diol (**56–57**, Figure 9), together with the isomers of **57**, which are (2*R*,3*R*)-butane-2,3-diol and (2*S*,3*S*)-butane-2,3-diol (**58–59**, Figure 9), although no phytotoxic activity has yet been determined [58,59]. Recently, a bioactive disubstituted nonenolide, named truncatenolide (**60**), and a new trisubstituted oct-2-en-4-one, named truncatenone (**61**), and the well-known tyrosol and *N*-acetyltyramine (**9** and **62**) have also been described (Figure 9). 

Truncatenolide (**60**) showed the strongest phytotoxic activity when tested on soybean seeds while tyrosol and *N*-acetyltyramine (**9** and **62**) exhibited phytotoxicity to a lesser extent. Furthermore, truncatenone (**61**) weakly stimulated the growth of the seed root in the condition tested [60]. When the same metabolites were assayed against *M. phaseolina* and *C. nicotianae*, truncatenolide (**60**) showed significant antifungal activity against *M. phaseolina* and the total inhibition of *C. nicotianae*. Thus, some other fungal nonenolides and their derivatives were assayed for their antifungal activity against both fungi in comparison with truncatenolide for a structure–activity relationship study.

Lupindolinone, lupinlactone, (3*R*)-mevalonolactone, and tyrosol (**63–65** and **9**, Figure 10) were isolated from *Colletotrichum lupini*, which is the causal agent of anthracnose in lupin (*Lupinus albus*) [61]. When these metabolites were tested for their toxicity through different experiments including the effect on root elongation in cress (*Nasturtium officinale*), lupine and duckweed (*Lemma minor*) leaves, or on the seed germination of parasitic plants such as broomrape (*Phelipanche ramosa*), only lupinlactone (**64**) and tyrosol (**9**) showed the greatest activity out of all of them [61].

Colletotrichin and colletopyrone (**66** and **67**, Figure 11) have been isolated from *Colletotrichum lindemuthianum*, the causal agent of anthracnose on common bean (*Phaselous vulgaris*). The exudate filtrates from the fungal culture have been shown to cause necrotic spots on common bean leaves [114] and to inhibit the seed germination of cowpea (*Vigna unguiculata*), soybean (*Glycine max*), maize (*Zea mays*), sorghum (*Sorghum* spp.), and millet (*Panicum miliaceum*) [62].

#### Structure–toxicity relationship studies of phytotoxins from *Colletotrichum*

A structure–activity relationship study was carried out using truncatenolide and pinolidoxin, 7-*epi*-pinolidoxin, 7,8-*O*,*O*′-diacetylpinolidoxin [42], stagonolide C [115], modiolide A, and stagonolide H [116]. The last three nonenolides were obtained from *Stagonospora cirsii* and were previously proposed as a mycoherbicide to the biocontrol of *Cirsium arvense* and *Sonchus arvensis*, which are two common weeds limiting the growth of several cereal cultures. Among all of the tested nonenolides, pinolidoxin (**11**) showed low antifungal activity against both fungi, while modiolide A selectively and totally inhibited only the growth of *C. nicotianae*. These results show that their activity could be linked to the nonenolide ring [60].

### 2.5. Phytotoxic Compounds Produced by Cercospora *spp.*

*Cercospora* is a genus of fungi that causes pink-violet spots on the seeds and spreads as the plant develops [117], penetrating through the stomata of the leaf surface and colonizing the intercellular spaces. Initially, the necrotic red-violet lesions mainly affect the leaves, expanding rapidly to coalesce with adjacent lesions, resulting in severe blighting of the leaves, and conidia protrude in fasciculate bundles in moist conditions from the center. The symptoms were often confused with those developed from the fungi of the genus *Ascochyta* [118]. The spores can be dispersed by environmental agents such as rain or wind, although these can flourish in later or nearby crops if the infected remains are not removed. Environmental conditions such as high humidity and warm temperature are required for spore germination and fungal development [117]. Crop rotation, the usage of resistant varieties, and seeds treated to suppress spore development are useful practices applied to control the pathogen.

Cercosporin (**68**, Figure 12) is the only compound elucidated by *Cercospora kikuchii* [119]. Cercosporin (**68**) is a non-specific phytotoxin, which is reported to have a role in the pathogenicity of the fungus, as it has been tested on different hosts (such as *Ricinus communis* or *Phaseolus vulgaris* among others), causing chlorosis and necrosis in most of them [63]. Additionally, the production of cercosporin (**68**) varied between different species or strains, being produced through a polyketide pathway and regulated by the calcium/calmodulin complex or mitogen-activated protein kinase signaling pathways (MAPKs). Likewise, this production is also affected by many other physiological and environmental factors such as the availability of nutrients, the ratio between C:N, and the amount of light or temperature [120].

#### Structure–toxicity relationship studies of phytotoxins from *Cercospora* spp.

Cercosporin (**68**) has been classified as a photosensitizer, since the phytotoxicity of this compound depends on the intensity of light. This compound reacts with light, producing free radicals and active oxygen species, particularly singlet oxygen. These reactive compounds are those that induce degradation in the acids of the cell wall of the host, thus increasing the virulence of the disease [121].

### 2.6. Phytotoxic Compounds Produced by Fungi Pleiochaeta setosa

Brown spot disease induced by *Pleiochaeta setosa* can affect legumes in general, although its most common host is lupine (*L. albus*), producing a disease known as brown spot disease, characterized by the appearance of brown spots on the aerial parts of plants such as the leaves and stems and can even affect the root, causing leaf necrosis and finally total wilt [122]. Although there are chemical methods of control including fungicide application, their use has not been proven effective yet. Instead, there are effective physical methods such as the use of heat and low humidity to sterilize seeds [123]. 

Setosol (**69**, Figure 13) is the only compound produced by *P. setosa*, although its triacetylated derivative (**70**, Figure 13) has also been isolated in lesser quantity [64]. Setosol (**69**) was tested on four lupine variants against an unpurified fungal extract of *P. setosa* and it was observed that the same lesions occurred as an infection caused by the fungus, which led to the conclusion that this is the compound responsible for pathogenicity in the host [64]. However, when setosol is acetylated (**70**), the molecule significantly loses its effectiveness [124]. 

#### Structure–toxicity relationship studies of phytotoxins from *Pleiochaeta* spp.

It has been shown that the toxicity of setosol (**69**) is related to the hydroxyl groups of the carbons in positions 6, 10, and 11. Setosol (**69**) has been shown to be an unstable molecule, so natural acetylation increases its stability. However, as this acetylation increases, its phytotoxic activity decreases. After studying the phytotoxicity of these compounds and their structure, it has been hypothesized that the introduction of chlorine and bromine at these sites may increase the activity of the molecule [124]. 

### 2.7. Phytotoxic Compounds Produced by Sclerotinia *spp.*

*Sclerotinia* disease can cause serious yield loss and seed quality problems. This genus of fungi is characterized by the formation of an apothecium in which ascospores are formed, thus differing in the regulation of sexual reproduction [125]. Symptoms are similar to those induced by *Botrytis cinerea*, starting with white and hairy mycelium developing in the aerial parts of the plant that later darken and harden, being more common during the inflorescence period. Later, when the wilted parts fall to the ground or are handled during the farming operations, the spores spread through the soil, favoring the beginning of a new infection cycle [126]. The effective control of this disease depends on various factors such as irrigation, avoiding an excess of water, and the application of fungicides, which are more effective when applied in the full bloom of primary inflorescences. Among the tested fungicides, benomyl, thiophanate methyl, and vinclozolin prevented the appearance of symptoms in leaf tissue on greenhouse-grown soybean plants. Furthermore, vinclozolin was also effective in reducing the mycelium growth of the pathogen *Botrytis cinerea* when added to PDA culture medium [127,128].

Ethanedioic acid, better known as oxalic acid (**71**, Figure 14), is the only metabolite reported for *Sclerotinia sclerotiorum* [65]. It acts during the pathogenesis process of the fungus by breaking the host cell wall, maximizing the efficacy of the different enzymes produced by the pathogen [129]. Its cytotoxic activity has been tested on sunflower and tomato, showing its involvement during the process, but its role against other crops still needs to be evaluated [130].

In studies related to oxalic acid (71), it has been observed that it is a determining compound in the pathogenicity of the *Sclerotinia* fungus, seeing that increasing production of this compound causes greater damage to the host [65]. Likewise, it has been seen that this compound is metabolized with oxygen and carbon dioxide from the medium at the time of pathogenicity. However, the pathogenicity of oxalic acid (72), generated through the glyoxylate acid pathway [131], is closely related to its ability to manipulate enzymes involved in the plant defense mechanism. Alteration of these processes reduces the pathogenicity because the fungus is not able to extract nutrients involved in plant colonization [132]. 

## 3. Potential Application of Phytotoxins Isolated from Legume Fungal Pathogens

Some of the metabolites here reviewed have been shown to possess various biological activities including antibiotic, antifungal, antiviral, but also herbicidal activity. For this reason, they have been tested in other fields, and some of them possess valuable alternative uses, as summarized in Table 2.

### 3.1. Pharmacological Activity

Phytotoxins have been used for human benefit, playing an important role in the field of medicine, especially in forensic medicine, clinical toxicology, pharmacy, pharmacology, and veterinary medicines [59]. Most phytotoxins are known to be poisonous and toxic to humans [144]. However, some of them display antimicrobial and antitumor activity, which can be used in drug development and new discoveries. Phytotoxins play a vital role in cell-cycle regulation, DNA disruption, cytotoxicity, and anticancer effects. Several phytochemicals have been proposed as potential antimicrobial and antitumor agents that could serve as alternatives to traditional medicine and have been investigated as potential antibiotics due to the current problem of the appearance of strains resistant to common antibiotics. One of these cases studied was tyrosol (**9**), which has been shown to have antioxidant properties and to act as an antimicrobial agent against *Staphylococcus aureus* and *Escherichia coli* [134], and it has also been shown to be a quorum sensing molecule of *Candida albicans* [145]. In the case of ascochitine (**19**), it has been shown to have antibiotic activity against certain lines of fungi and bacteria such as *S. aureus* or *C. albicans*, although this type of study needs to be further developed in this field [136]. 

In the same way, the search for antitumor agents is on the rise in an attempt to avoid more invasive techniques that harm the state of health. One of these cases involves the metabolite pseurotin A (**10**), which after carrying out in vitro and in vivo tests, was shown to act as an antitumor agent. [129]. Concerning cercosporin (**68**), it has been shown that it can act as a photosensitizer for the activation of certain drugs in therapy against various types of tumors [140].

Finally, with respect to pinolidoxin (**11**), other studies have been carried out to assist in further research. This compound caused a clearly detectable actin microfilament disruption in the NIH/3T3 fibroblast cells, being less toxic than its homologue latrunculin A (which is the most commonly used drug in this area), and also accommodates significant structural changes without engendering a loss of bioactivity [146], thus improving the experiments carried out for the study of the cytoskeleton.

### 3.2. Crop Protection

A great amount of work has been conducted to test and describe the utility of phytotoxin applications in agriculture, enhancing crops development as (1) natural herbicidal agents, and (2) pest or disease management. In fact, phytotoxins can be effectively used as weed control agents due to characteristics such as selectivity, low persistence in the environment, and environmental safety compared with synthetic herbicides [147]. The huge structural variety and advanced biological activity of these natural phytotoxins make them promising candidates for the development of natural herbicides that use new modes of action in both natural and well-controlled environments. One of the most important features is that they can aim at new sites of action, thereby reducing any form of herbicide resistance. 

Phytotoxins produced by plant pathogens can be extracted both from damaged plant tissues or from an artificial culture medium and then used as defensive agents in other crops against a variety of viruses, bacteria, fungi, and insects. Lentiquinone A (**1**) recently showed some fungicidal activity against certain fungi such as *Verticillium dahlia*, *Penicillium allii*, *Rhizoctonia* spp., and *Phoma exigua* [72]. Lentiquinone B, lentiquinone C, and lentisone (**2–4**) exhibited antibiotic properties against the bacteria *Bacillus subtilis*, lentiquinone B (**2**) being the one with the lowest proportion of antibiosis of the three [72]. Pachybasin (**8**) and lentiquinone C (**3**), isolated from *A. lentis*, have been tested against rust and powdery mildew fungi of legumes and cereals of agronomic importance and showed significant preventive and curative effects [132]. Pinolidoxin (**11**) selectively inhibits cell growth (tested in vitro in a culture of cells extracted from a *Populus trichocarpa* canker), inhibiting phenylalanine ammonia-lyase, which suggests that it can possibly inhibit the defense system of the host plant [79]. This compound was also tested to verify its potential as a fungicide and it was shown that it is capable of inhibiting the growth of *C. nicotianae* in the same way as truncatenolide (**60**) [60].

Lathyroxin A and Lathyroxin B (**20** and **21**) inhibited the germination of the seed of the parasitic plant *Phelipanche ramosa* [136]. Regiolone (**29**) inhibited pathogens such as *Bacillus subtilis*, *Colletotrichum gloeosporioides*, and *Magnaporthe oryzae* [139]. Botrydial (**35**) can act against bacteria of the genus *Bacillus* [140], while botrylactone (**47**) also has antibacterial activity against *Bacillus mycoides* and *Bacillus subtilis* [141].

Solanapyrone A (**24**) has fungicidal activity, since it is capable of inhibiting the competitors of *A. rabiei* such as other fungi of the genus *Alternaria*, *Epicoccum*, or *Ulocladium* when the fungus has been established and has begun its infective process [139]. Setosol (**69**) mainly has two types of activities: one as a fungicide against fungi from families such as *Colletotrichum*, *Drechslera*, *Gerlachia*, and *Pyricularia* and also acts against yeasts of the genus *Cryptococcus* and against bacteria of the genus *Staphylococcus* [124].

Oxalic acid (**71**) is proposed as a pesticide against varroa mites (*Varroa destructor*), which is a very widespread parasite of honeybees (*Apis mellifera*) throughout the world, being able to eliminate the parasite from a whole hive in few months [148]. Oxalic acid (**71**) is also effective in periods without rearing, but has a very limited effect when the parasite is in its rearing period [143].

## 4. General Conclusions and Perspectives

Necrotrophic fungi are pathogens responsible for a wide variety of severe fungal diseases that can cause enormous losses in crops, and therefore cause great economic losses [149]. These fungi, among the different compounds that they are capable of secreting, produce phytotoxins, which have been determined to play a closely related role in the plant infection process, and only approximately 25% of these secondary compounds have been characterized in fungi [150]. Although only phytotoxins from necrotrophic fungi have been discussed in this review, it is also interesting to study and broaden the knowledge of other secondary metabolites that fungi are capable of exuding, since they also participate in cell regulation and in the development of the disease by modifying the metabolism of the plant.

Although a great variety of compounds produced by different fungi have been described over the years, this research field is still thriving. Therefore, for future projects, one of the main perspectives is to expand the knowledge of these fungi, investigating the metabolic route for secondary metabolite production [151]. In addition, multidisciplinary investigation groups worldwide have started to work on secondary metabolites produced by less common fungal species such as the necrotrophic fungus *Stemphylium*. Indeed, a recent metabolomic study was carried out by identifying already known metabolites; nevertheless, several features remain unknown due to the absence of satisfactory matches in the databases. Moreover, the complete biological characterization of the identified compound has not yet been carried out [152]. 

Fungal phytotoxins have potential applications in chemotaxonomy due to their species-specific production. The presence or absence of these compounds can be used to distinguish between closely related fungal species [153]. Indeed, this can be combined with molecular biology techniques such as DNA sequencing and fingerprinting, enhancing the accuracy and reliability of chemotaxonomic studies. This approach can also provide insights into the evolution and ecology of fungal species.

Studying the fungal secondary metabolites can also be useful for investigating plant–pathogen interactions. Several analytical techniques that allow a little manipulation and sample preparation have been developed for this kind of study. One of the most advanced analytical methods is desorption electrospray ionization mass spectrometry (DESI-MS). The technique is particularly useful for studying plant–pathogen interactions as it allows for the detection of a wide range of metabolites that could be involved in the interaction. The detectable metabolites include both volatile compounds that are emitted by plants in response to infection and non-volatile compounds such as lipids and proteins, which can provide information about the structural changes that occur in the plant because of infection. Another significant application of DESI-MS in plant–pathogen interaction is its ability to identify specific metabolites produced by the pathogen, favoring the chemotaxonomy [154,155]. The results of these studies might be employed to develop more effective strategies for controlling the pathogen and protecting the plant. 

Indeed, fungal phytotoxins have the potential to serve as biomarkers for plant diseases due to their specificity. Recent studies have demonstrated that phytotoxins are produced during infection and can be detected in plant tissues. For instance, studies have been carried out for Fusarium head blight in wheat and corn or for Alternaria leaf blight [156,157]. They are ideal candidates for early disease diagnosis, and recent advances in techniques such as the already cited mass spectrometry or ELISA assay, further support their potential use as reliable and cost-effective markers. Moreover, investigating fungal phytotoxins could also be helpful in biocontrol. There is a worldwide paradigm shift toward reducing the usage of chemical pesticides. In this framework, fungal antagonists play a significant role as biocontrol agents (BAs) [158]. Thus, investigating phytotoxins, or more in general, bioactive secondary metabolites, can also help understand the BA–pathogen interaction. Co-culture studies applying high throughput screening techniques have already highlighted how the microbial community interacts [159]. However, extensive in planta studies in controlled conditions or field studies have yet to be reported. Understanding the role of fungal phytotoxins in interacting with other life forms is critical for developing effective disease management strategies. Subsequently, following complete chemical and biological characterization including ecotoxicological studies, phytotoxins could be useful lead compounds in several scientific fields, either in medicine [144] or in agriculture [132,160]. Indeed, they could be applied to the crop as a spray or incorporated into the soil to provide long-term protection, being a safe and effective alternative to traditional chemical pesticides and can be used to reduce the risk of resistance development and environmental pollution.

In the past decades, metabolomics has emerged as a powerful tool for studying fungal phytotoxins. Targeted and non-targeted metabolomics approaches have been used to identify and quantify bioactive known secondary metabolites including phytotoxins. Metabolomics can help identify the metabolic changes in plants following exposure to fungal phytotoxins and understand the mechanisms by which these compounds cause disease symptoms [161]. Moreover, it can be used in combination with genomics and transcriptomics. For instance, combining metabolomics and genomics can provide insights into the genetic basis of phytotoxin production. Similarly, its integration with transcriptomics might reveal the regulatory mechanisms that govern phytotoxin biosynthesis and help to identify potential molecular targets in the host plants. Nevertheless, there is still room for improvement, particularly in protocols and method standardization [162].

Phytotoxins are also used in strategies finalized to induce resistance against fungal pathogen producers and responsible for the disease. The use of resistance inducers represents a helpful alternative for management of the plant disease, which can give good results when included in integrated biocontrol programs. Some examples of satisfactory fungal disease control achieved by the use of phytotoxins produced by pathogens have already been described. Thaxtomin A, the main phytotoxin produced by *Phakopsora pachyrhizi*, was successfully used in the induction of resistance of Asian soybean rust (ASR), caused by this pathogen [163]. *Magnaporthe grisea* is a ubiquitous fungus responsible for finger millet blast, the most devastating disease affecting this cereal. Their fungal toxins have been useful for the development of resistant varieties and their screening procedures [164]. In recent years, tissue culture based on in vitro selection has emerged as a feasible and cost-effective tool for developing stress-tolerant plants. Plants tolerant to biotic stresses can be acquired by applying selecting agents such as pathogen culture filtrate, phytotoxins, or the pathogen itself (for disease resistance) in the culture [165].

Finally, due to changes in environmental conditions, mainly the result of climate change, fungal pathogens are expanding their host range [166]. The World Health Organization (WHO) has observed an increase in fungal diseases in which humans act as a transmission vector. The spread of fungal diseases is mainly harmful to immunocompromised patients. The diagnosis of this type of disease is complicated due to the scarcity of data in medicine. Additionally, their treatment is complicated by the absence of antifungal drugs [167]. For this reason, the WHO has recently proposed lines of action for their control and study, starting from regional surveillance activities to an improvement in its diagnosis. Finally, the commission also suggested the development of a multidisciplinary platform that facilitates the transfer of knowledge between different investigation fields to facilitate the understanding of pathophysiology for the development of antifungal drugs for the treatment of this type of disease [167]. Therefore, the experience of plant pathologists, biologists, and chemists who have worked in the frame of plant protection might be fundamental to develop scientific projects with a One-Health perspective.

## Figures and Tables

**Figure 1 ijms-24-05116-f001:**
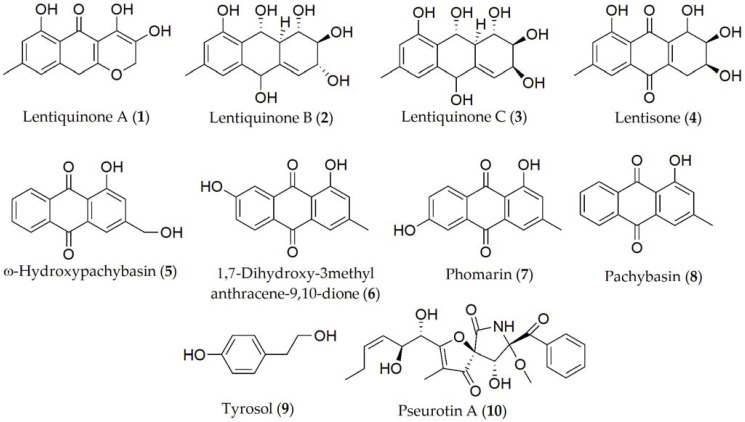
Phytotoxins isolated from *Ascochyta lentis*.

**Figure 2 ijms-24-05116-f002:**
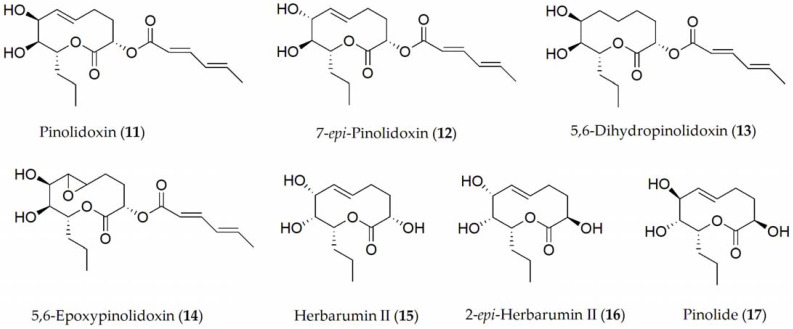
Phytotoxins isolated from *Ascochyta pinodes*.

**Figure 3 ijms-24-05116-f003:**
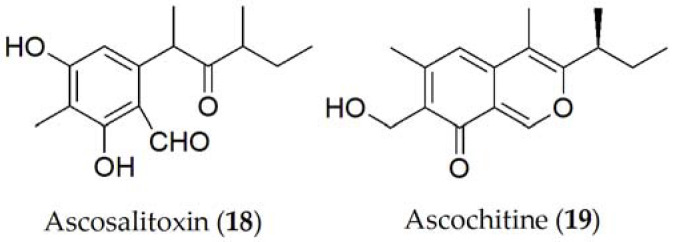
Phytotoxins isolated from *Ascochyta pisi*.

**Figure 4 ijms-24-05116-f004:**
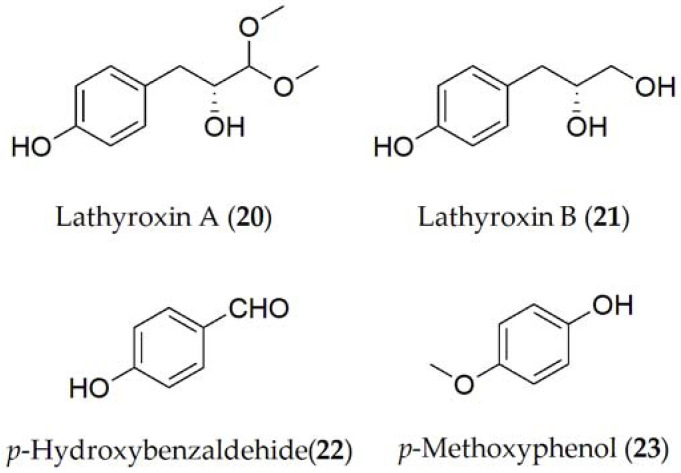
Phytotoxins isolated from *Ascochyta pisi* var. *lathyri*.

**Figure 5 ijms-24-05116-f005:**
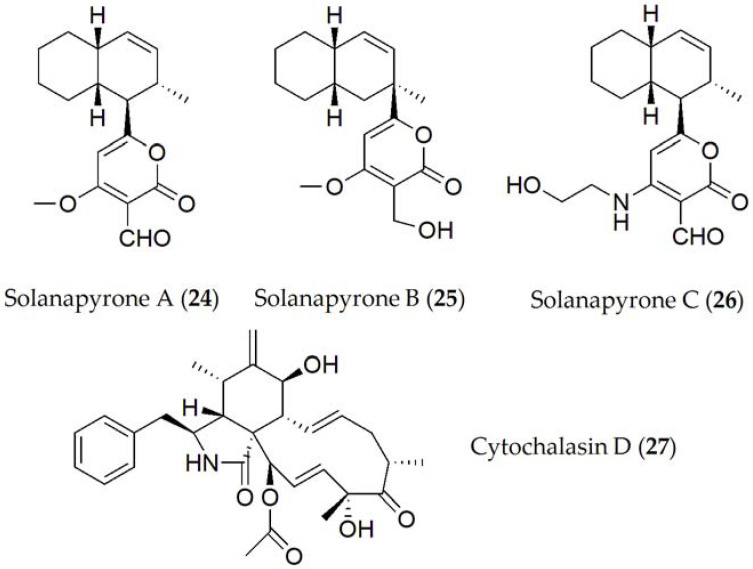
Phytotoxins isolated from *Ascochyta rabiei*.

**Figure 6 ijms-24-05116-f006:**
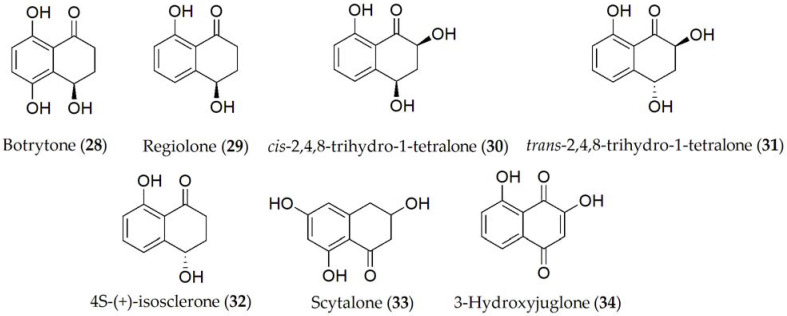
Phytotoxins isolated from *Botrytis fabae*.

**Figure 7 ijms-24-05116-f007:**
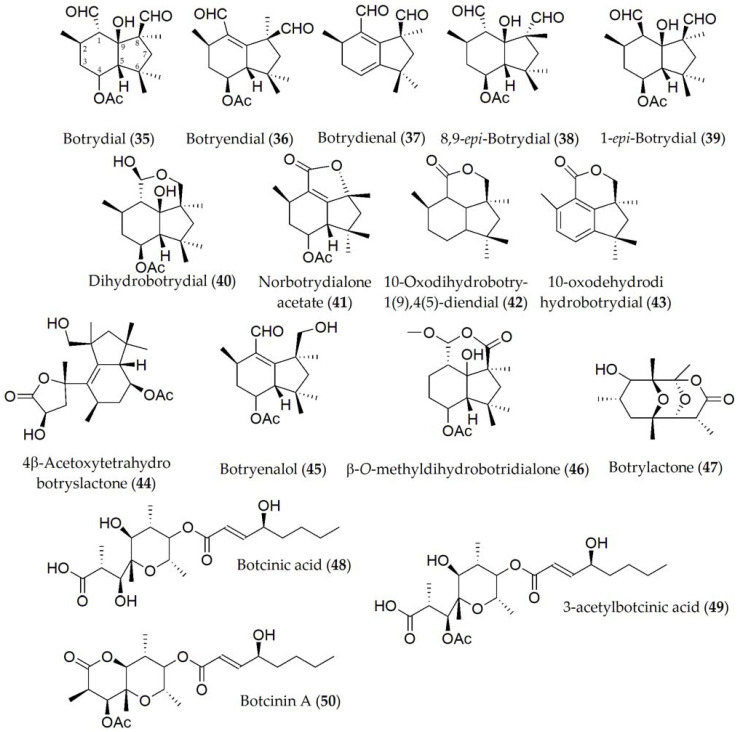
Phytotoxins isolated from *Botrytis cinerea*.

**Figure 8 ijms-24-05116-f008:**
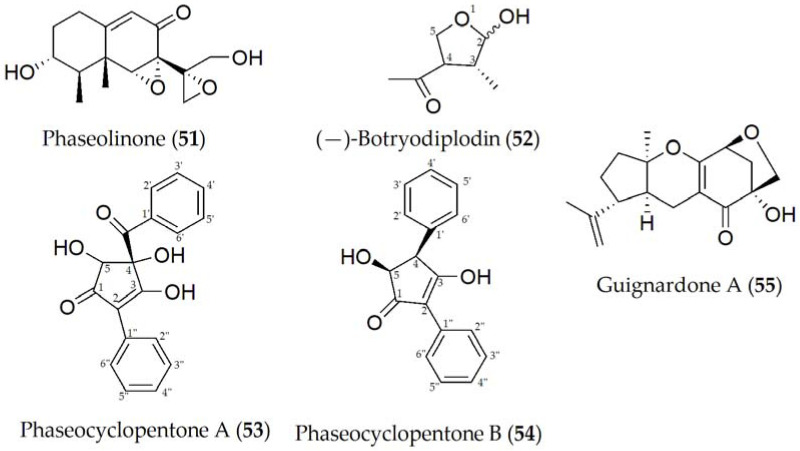
Phytotoxins isolated from *Macrophomina phaseolina*.

**Figure 9 ijms-24-05116-f009:**
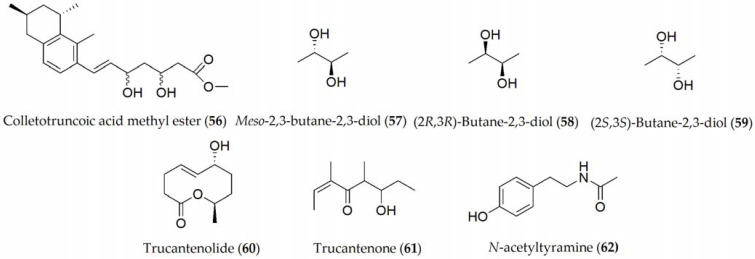
Phytotoxins isolated from *Colletotrichum truncatum*.

**Figure 10 ijms-24-05116-f010:**
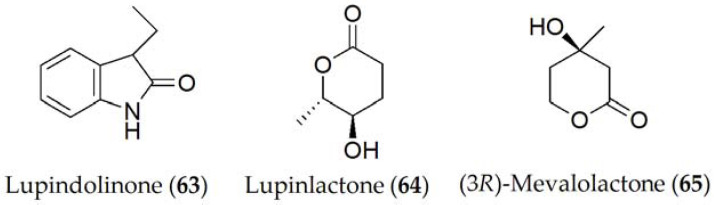
Phytotoxins isolated from *Colletotrichum lupini*.

**Figure 11 ijms-24-05116-f011:**
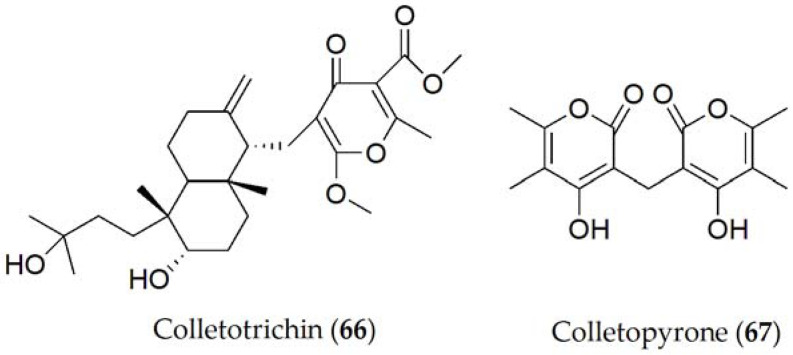
Phytotoxins isolated from *Colletotrichum lindemuthianum*.

**Figure 12 ijms-24-05116-f012:**
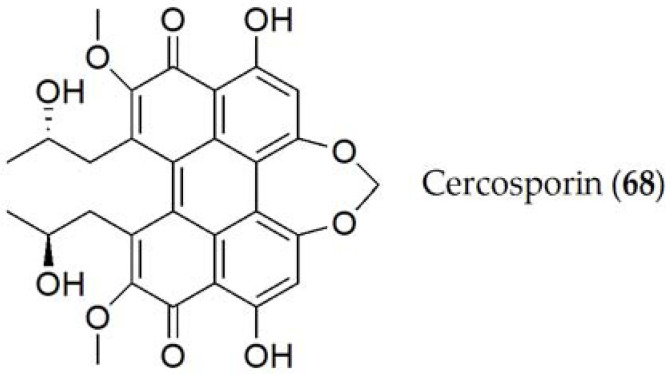
Phytotoxins isolated from *Cercospora kikuchii*.

**Figure 13 ijms-24-05116-f013:**
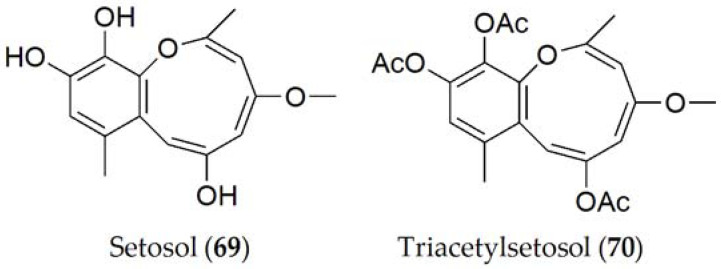
Phytotoxins isolated from *Pleiochaeta setosa*.

**Figure 14 ijms-24-05116-f014:**
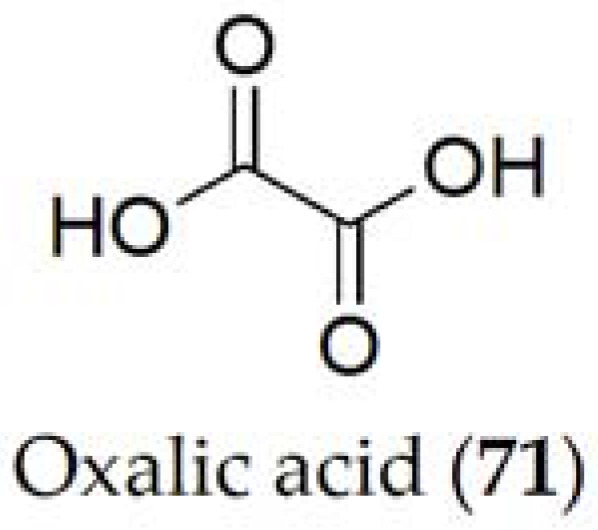
Phytotoxin isolated from *Sclerotinia sclerotiorum*.

**Table 1 ijms-24-05116-t001:** Metabolites reported on necrotic fungi infecting grain legumes.

Fungi	Host	Disease	Metabolite	References
*Ascochyta lentis*	Lentil (*Lens culinaris*)	Ascochyta blight	Lentiquinone A (**1**)	[21]
			Lentiquinone B (**2**)	[21]
			Lentiquinone C (**3**)	[21]
			Lentisone (**4**)	[41]
			ω–Hydroxypachybasin (**5**)	[21]
			1,7-Dihydroxy-3-methylan thracene-9,10-dione (**6**)	[21]
			Phomarin (**7**)	[21]
			Pachybasin (**8**)	[41]
			Tyrosol (**9**)	[41]
			Pseurotin A (**10**)	[41]
*Ascochyta pinodes*	Pea (*Pisum sativum*)	Ascochyta blight	Pinolidoxin **(11**)	[42]
			7-*epi*-Pinolidoxin (**12**)	[43,44]
			5,6-Dihydropinolidoxin (**13**)	[43,44]
			5,6-Epoxypinolidoxin (**14**)	[43,44]
			Herbarumin II (**15**)	[44]
			2-*epi*-Herbarumin II (**16**)	[44]
			Pinolide (**17**)	[44]
*Ascochyta pisi*	Pea (*Pisum sativum*)	Ascochyta blight	Ascosalitoxin (**18**)	[45]
			Ascochitine (**19**)	[42]
*Ascochyta lentis* var. *lathyri*	Grass pea (*Lathyrus sativus*)	Ascochyta blight	Lathyroxin A (**20**)	[22]
			Lathyroxin B (**21**)	[22]
			*p*-Hydroxybenzaldehyde (**22**)	[22]
			*p*-Methoxyphenol (**23**)	[22]
			Tyrosol (**9**)	[22]
*Ascochyta fabae*	Faba bean (*Vicia faba*)	Ascochyta blight	Ascochitine (**19**)	[46]
*Ascochyta rabiei*	Chickpea (*Cicer arietinum*)	Ascochyta blight	Solanapyrone A (**24**)	[47]
			Solanapyrone B (**25**)	[48]
			Solanapyrone C (**26**)	[47]
			Cytochalasin D (**27**)	[49]
*Botrytis fabae*	Faba bean (*Vicia faba*)	Chocolate spot	Botrytone (**28**)	[50]
			Regiolone (**29**)	[50]
			*cis*-2,4,8-Trihydroxy-1-tetralone (**30**)	[50]
			*trans*-2,4,8-Trihydroxy-1-tetralone (**31**)	[50]
			(4*S*)-(+)-Isosclerone (**32**)	[50]
			Scytalone (**33**)	[50]
			3-Hydroxyjuglone (**34**)	[50]
*Botrytis cinerea*	Faba bean (*Vicia faba*)	Grey mold	Botrydial (**35**)	[51]
			Botryendial (**36**)	[52]
			Botrydienal (**37**)	[52]
			8,9-*epi*-Botrydial (**38**)	[52]
			1-*epi*-Botrydial (**39**)	[52]
			Dihydrobotrydial (**40**)	[52]
			Norbotrydialone acetate (**41**)	[53]
			10-Oxodihydrobotry-1(9), 4(5)-diendial (**42**)	[53]
			10-Oxodehydrodihydro botrydial (**43**)	[53]
			4β-Acetoxytetrahydro botryslactone (**44**)	[53]
			Botryenalol (**45**)	[54]
			β-O-Methyldihydrobotri dialone (**46**)	[54]
			Botrylactone (**47**)	[54]
			Botcinic acid (**48**)	[54]
			3-Acetylbotcinic acid (**49**)	[54]
			Botcinin A (**50**)	[54]
*Macrophomina phaseolina*	Soybean (*Glycine max*)	Charcoal rot	Phaseolinone (**51**)	[55]
			(-)-Botryodiplodin (**52**)	[56]
			Phaseocyclopentenones A (**53**)	[57]
			Phaseocyclopentenones B (**54**)	[57]
			Guignardone A (**55**)	[57]
*Colletotrichum truncatum*	Soybean (*Glycine max*)	Anthracnose	Colletruncoic acid methyl ester (**56**)	[58,59]
			*Meso*-2,3-butane-2,3-diol (**57**)	[58,59]
			(2*R*,3*R*)-Butane-2,3-diol (**58**)	[58,59]
			(2*S*,3*S*)-Butane-2,3-diol (**59**)	[58,59]
			Truncatenolide (**60**)	[60]
			Truncatenone (**61**)	[60]
			*N*-Acetyltyramine (**62**)	[60]
			Tyrosol (**9**)	[60]
*Colletotrichum lupini*	Lupine (*Lupinus albus*)	Anthracnose	Lupindolinone (**63**)	[61]
			Lupinlactone (**64**)	[61]
			(3*R*)-Mevalonolactone (**65**)	[61]
			Tyrosol (**9**)	[61]
*Colletotrichum lindemuthianum*	Common bean (*Phaseolus vulgaris*)	Anthracnose	Colletotrichin (**66**)	[62]
			Colletopyrone (**67**)	[62]
*Cercospora kikuchii*	Soybean (*Glycine max*)	Pod and stem blight	Cercosporin (**68**)	[63]
*Pleiochaeta setosa*	Lupine (*Lupinus albus*)	Brown spot	Setosol (**69**)	[64]
			Triacetylsetosol (**70**)	[64]
*Sclerotinia sclerotiorum*	Common bean (*Phaseolus vulgaris*)	White mold	Oxalic acid (**71**)	[65]

**Table 2 ijms-24-05116-t002:** Discovered biological activities of some of the reported phytotoxins.

Metabolites	Biological Activity	Literature
Lentiquinone A (**1**)	Fungicidal activity	[72]
Lentiquinone B (**2**)	Antibiotic activity	[72]
Lentiquinone C (**3**)	Antibiotic activity	[72]
	Fungicidal activity	[133]
Lentisone (**4**)	Antibiotic activity	[21]
Pachybasin (**8**)	Fungicidal activity	[133]
Tyrosol (**9**)	Antioxidant effect, anti-bacterial activity	[134]
Pseurotin A (**10**)	Anti-tumoral activity	[135]
Pinolidoxin (**11**)	Cytotoxic activity	[79]
	Fungicidal activity	[60]
Ascochitine (**19**)	Antibiotic activity	[136]
Lathyroxin A (**20**)	Herbicidal activity	[137]
Lathyroxin B (**21**)	Herbicidal activity	[137]
Solanapyrone A (**24**)	Fungicidal activity	[138]
Regiolone (**29**)	Anti-bacterial activity Fungicidal activity	[139]
Botrydial (**35**)	Anti-bacterial activity	[140]
Botrylactone (**47**)	Anti-bacterial activity	[141]
Truncatenolide (**60**)	Fungicidal activity	[60]
Cercosporin (**68**)	Anti-tumoral activity	[142]
Setosol (**69**)	Anti-bacterial activity Fungicidal activity	[122]
Oxalic acid (**71**)	Pesticide	[143]

## Data Availability

The data presented in this study are available on request from the corresponding author.

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
