# Peer review of "Status of Phytotoxins Isolated from Necrotrophic Fungi Causing Diseases on Grain Legumes"

_ijms, 2023, doi:10.3390/ijms24065116_

Round 1
Reviewer 1 Report
Dear Author,
I reviewed this manuscript and found it to be considerbly well written and interesting.
In this review authors have reported the isolation, chemical and biological characterization of fungal phytotoxins produced by the most important necrotrophic
fungi involved in legumes diseases. In addition, they have discussed the mode of actions of the phytotoxins.
All the necessary changes have mentioned in the relevant places (see the attached).
I would like to see a minor revision of this manuscript.
Thank you.

Author Response
We thank reviewer 1 for the comments to our work and the valuable suggestions that will improve the quality of our publication. As you can see, all the suggestions have been now included into the text.
Reviewer 2 Report
This paper thoroughly and meticulously reviews the current state of the field of fungal pathogen-plant host interactions in terms of low molecular weight toxin production. In general, the paper is detailed, accurate, and comprehensive and I can find no major alterations that need to be made prior to publication. Nonetheless, a set of grammatical or clarifying changes would benefit the paper in terms of increasing its ‘smoothness’ and readability. Some suggestions avoid words with multiple meanings that could confuse a reader; others avoid colloquialisms. Note that comments in quotations are merely suggestions for smoother text; adjust as needed. A few suggestions include:
Line 4: Check the formatting on the Author list, specifically callouts for institution affiliation. For instance, “Agudo-Jurado, F.J.1, Reveglia, P.1…” etc.
Line 26-30: Consider splitting into 2 sentences. Also add punctuation.
Line 31: “The demands for protein crops are markedly increasing [7]; this should be paired…”
Line 32: “As with any crop, legumes can…”
Line 39: “…all fungi can be recognized by plant immune systems…”
Line 44-45: “When PAMPs are recognized through PRRs they trigger a fast…”
Line 63: “…inactivate the plant defenses by secreting…”
Line 64: “…enzymes, catalyzing the degradation…”
Line 68-70: “Induction of host alterations such as DNA damage, abnormal mitochondrial oxidation, cytotoxicity, etc. with the goal of leading to host cell death are some of the common functions…”
Line 74: “Although their true role is an open question, high molecular weight…”
Line 100: “…environmental adaptation aiming at stabilizing agricultural systems.”; I think this is the intended meaning
Line 101: “Herein, we review the isolation…”
Line 145-146: “By contrast, the other metabolites found only produce reduced…”
Line 148: “…did not display significant…”
Line 157: “…largely produced…” – abundantly produced? Please alter to clarify.
Line 158: “…(19, Figure 3), where it displayed…”
Line 161: p-hydroxybenzaldehyde? Check spelling.
Line 170: “Concerning A. lentis var. lathyri, the compounds described have…”
Line 182: “…solanapyrones A, B and C were shown to be…”
Line 183-4: “…being able to reduce root development as well as the seed germination…”
Line 189: “As an example…”
Line 192: “The two nonenolides…”
Line 211: “…unbalancing the cell cycle and finally causing apoptosis…”
Line 221-2: “…average temperature of around 22 °C, the fungus begins…”
Line 227: “…the seed quality decreases…”
Line 229-232: This sentence is somewhat confusing, please consider re-wording.
Line 233: “…which is why other strategies have been formulated…”
Line 249-50: “…extracted from other hosts, including legumes such as…”
Line 252: “Compounds identified in B. cinerea (table 1) include botrydial, botryendial…”
Line 268-9: “Botrydial (35) was the only phytotoxin produced by Botrytis cinerea for which SAR studies were conducted.”
Line 278-9: “…causing charcoal rot on soybean, although it can also affect other crop legumes such as cowpea or common bean…”
Line 301: “…may help to explain the highly efficient…”
Line 311-12: “…which are two severe pathogens both isolated from infected soybean plants…”
Line 325-6: “…substrate for the ribose 5-kinase enzyme, or it may exert another function when present in the cell cytoplasm in its phosphorylated form...”
Line 337: “…pathogens that can affect a multitude of hosts including…”
Line 353: “…named truncatenolide (60) and a new trisubstituted…”
Line 357-8: “…exhibited phytotoxicity to a lesser extent.”
Line 370: “…different experiments, including the effect on root elongation…”
Line 371: “…seed germination of parasitic plants such as…”
Line 383-8: Run-on sentence, consider splitting into two sentences.
Line 385: “…with the last three nonenolides…”
Line 401: “This fungus requires…”
Line 409: “…which is reported to have a role…”
Line 429-30: “Although there are chemical methods of control including fungicide application, their use has not yet been proven effective.”
Line 435: “…is the only compound produced by P. setosa…”
Line 440: “…when setosol is acetylated (70), the molecule loses its…”
Line 443: “…shown that the toxicity of setosol…”; I think this is the meaning.
Line 444: “Setosol (69) has been shown to be…”
Line 462-4: The meaning of this sentence is not completely clear; please try re-wording.
Line 480: “…because the fungus is not able to extract…”
Line 483-4: “Some of the metabolites here reviewed have been shown to possess various biological activities including antibiotic, antifungal, antiviral, but also herbicidal activity.”
Line 497-8: “…alternatives to traditional medicine and have been investigated as potential antibiotics due to…”
Line 499-500: “…One of these cases studied was tyrosol (9), which has been shown to have antioxidant properties and to act as an antimicrobial agent…”
Line 501: “…it has also been shown to be a quorum sensing…”
Line 505-6: “In the same way, the search for antitumor agents is on the rise in an attempt to avoid more invasive techniques…”
Line 507-8: “…carrying out in vitro and in vivo tests, it was shown to can act as an antitumor agent.”
Line 510: “…therapy against various types of tumors…”
Line 512-15: “…This compound caused a clearly detectable actin microfilament disruption in NIH/3T3 fibroblast cells, being less toxic than its homologue latrunculin A (which is the most commonly used drug in this area)…”
Line 514-15: Not sure what is meant by “a better edition in the structure”; please consider rephrasing.
Line 530: “…against a variety of viruses, bacteria…”
Line 534: “…bacteria Bacillus subtilis, lentiquinone B (2) being the one…”
Line 538: “Pinolidoxin (11) selectively inhibits cell growth…”
Line 548: Bacillus mycoides?
Line 565: “…secreting, produce phytotoxins…”
Line 567-70: “Although only phytotoxins from necrotrophic fungi have been discussed in this review, it is also interesting to study and broaden the knowledge of other secondary metabolites that fungi are capable of exuding…”
Line 573: “…described over the years, this field…”
Line 575: “…secondary metabolite production…”
Line 580: “…due to the absence of satisfactory matches…”
Line 597: “Subsequently, following the complete chemical and biological…”
Line 603-4: “Finally, due to changes in environmental conditions, mainly the result of climate change, fungal pathogens…”
Line 615-16: “…biologists and chemists who have worked in the field of plant protection…”
Author Response
This paper thoroughly and meticulously reviews the current state of the field of fungal pathogen-plant host interactions in terms of low molecular weight toxin production. In general, the paper is detailed, accurate, and comprehensive and I can find no major alterations that need to be made prior to publication.
Answer: We thank the reviewer for these positive comments on our work
Nonetheless, a set of grammatical or clarifying changes would benefit the paper in terms of increasing its ‘smoothness’ and readability. Some suggestions avoid words with multiple meanings that could confuse a reader; others avoid colloquialisms. Note that comments in quotations are merely suggestions for smoother text; adjust as needed.
Answer: Thank you for your valuable suggestions that will improve the quality of our publication. As you can see, most of the suggestions have been now included into the text.
A few suggestions include:
Line 4: Check the formatting on the Author list, specifically callouts for institution affiliation. For instance, “Agudo-Jurado, F.J.1, Reveglia, P.1…” etc.
Answer: ok
Line 26-30: Consider splitting into 2 sentences. Also add punctuation.
Answer: The sentence have not been splitted, but have been rephrased and punctuation has also been added.
Line 31: “The demands for protein crops are markedly increasing [7]; this should be paired…”
Answer: ok
Line 32: “As with any crop, legumes can…”
Answer: ok
Line 39: “…all fungi can be recognized by plant immune systems…”
Answer: ok
Line 44-45: “When PAMPs are recognized through PRRs they trigger a fast…”
Answer: ok
Line 63: “…inactivate the plant defenses by secreting…”
Answer: ok
Line 64: “…enzymes, catalyzing the degradation…”
Answer: ok
Line 68-70: “Induction of host alterations such as DNA damage, abnormal mitochondrial oxidation, cytotoxicity, etc. with the goal of leading to host cell death are some of the common functions…”
Answer: ok
Line 74: “Although their true role is an open question, high molecular weight…”
Answer: ok
Line 100: “…environmental adaptation aiming at stabilizing agricultural systems.”; I think this is the intended meaning
Answer: ok
Line 101: “Herein, we review the isolation…”
Answer: ok
Line 145-146: “By contrast, the other metabolites found only produce reduced…”
Answer: ok
Line 148: “…did not display significant…”
Answer: ok
Line 157: “…largely produced…” – abundantly produced? Please alter to clarify.
Answer: ok
Line 158: “…(19, Figure 3), where it displayed…”
Answer: ok
Line 161: p-hydroxybenzaldehyde? Check spelling.
Answer: p-hydroxybenzaldehyde (para-hydr…) or 4- hydroxybenzaldehyde is the same. The compound’s name is correct.
Line 170: “Concerning A. lentis var. lathyri, the compounds described have…”
Answer: ok
Line 182: “…solanapyrones A, B and C were shown to be…”
Answer: ok
Line 183-4: “…being able to reduce root development as well as the seed germination…”
Answer: ok
Line 189: “As an example…”
Answer: ok
Line 192: “The two nonenolides…”
Answer: ok
Line 211: “…unbalancing the cell cycle and finally causing apoptosis…”
Answer: ok
Line 221-2: “…average temperature of around 22 °C, the fungus begins…”
Answer: ok
Line 227: “…the seed quality decreases…”
Answer: ok
Line 229-232: This sentence is somewhat confusing, please consider re-wording.
Answer: ok
Line 233: “…which is why other strategies have been formulated…”
Answer: ok
Line 249-50: “…extracted from other hosts, including legumes such as…”
Answer: ok
Line 252: “Compounds identified in B. cinerea (table 1) include botrydial, botryendial…”
Answer: ok
Line 268-9: “Botrydial (35) was the only phytotoxin produced by Botrytis cinerea for which SAR studies were conducted.”
Answer: ok
Line 278-9: “…causing charcoal rot on soybean, although it can also affect other crop legumes such as cowpea or common bean…”
Answer: ok
Line 301: “…may help to explain the highly efficient…”
Answer: ok
Line 311-12: “…which are two severe pathogens both isolated from infected soybean plants…”
Answer: ok
Line 325-6: “…substrate for the ribose 5-kinase enzyme, or it may exert another function when present in the cell cytoplasm in its phosphorylated form...”
Answer: ok
Line 337: “…pathogens that can affect a multitude of hosts including…”
Answer: ok
Line 353: “…named truncatenolide (60) and a new trisubstituted…”
Answer: ok
Line 357-8: “…exhibited phytotoxicity to a lesser extent.”
Answer: ok
Line 370: “…different experiments, including the effect on root elongation…”
Answer: ok
Line 371: “…seed germination of parasitic plants such as…”
Answer: ok
Line 383-8: Run-on sentence, consider splitting into two sentences.
Answer: ok
Line 385: “…with the last three nonenolides…”
Answer: ok
Line 401: “This fungus requires…”
Answer: ok
Line 409: “…which is reported to have a role…”
Answer: ok
Line 429-30: “Although there are chemical methods of control including fungicide application, their use has not yet been proven effective.”
Answer: ok
Line 435: “…is the only compound produced by P. setosa…”
Answer: ok
Line 440: “…when setosol is acetylated (70), the molecule loses its…”
Answer: ok
Line 443: “…shown that the toxicity of setosol…”; I think this is the meaning.
Answer: ok
Line 444: “Setosol (69) has been shown to be…”
Answer: ok
Line 462-4: The meaning of this sentence is not completely clear; please try re-wording.
Answer: ok
Line 480: “…because the fungus is not able to extract…”
Answer: ok
Line 483-4: “Some of the metabolites here reviewed have been shown to possess various biological activities including antibiotic, antifungal, antiviral, but also herbicidal activity.”
Answer: ok
Line 497-8: “…alternatives to traditional medicine and have been investigated as potential antibiotics due to…”
Answer: ok
Line 499-500: “…One of these cases studied was tyrosol (9), which has been shown to have antioxidant properties and to act as an antimicrobial agent…”
Answer: ok
Line 501: “…it has also been shown to be a quorum sensing…”
Answer: ok
Line 505-6: “In the same way, the search for antitumor agents is on the rise in an attempt to avoid more invasive techniques…”
Answer: ok
Line 507-8: “…carrying out in vitro and in vivo tests, it was shown to can act as an antitumor agent.”
Answer: ok
Line 510: “…therapy against various types of tumors…”
Answer: ok
Line 512-15: “…This compound caused a clearly detectable actin microfilament disruption in NIH/3T3 fibroblast cells, being less toxic than its homologue latrunculin A (which is the most commonly used drug in this area)…”
Answer: ok
Line 514-15: Not sure what is meant by “a better edition in the structure”; please consider rephrasing.
Answer: ok
Line 530: “…against a variety of viruses, bacteria…”
Answer: ok
Line 534: “…bacteria Bacillus subtilis, lentiquinone B (2) being the one…”
Answer: ok
Line 538: “Pinolidoxin (11) selectively inhibits cell growth…”
Answer: ok
Line 548: Bacillus mycoides?
Answer: yes
Line 565: “…secreting, produce phytotoxins…”
Answer: ok
Line 567-70: “Although only phytotoxins from necrotrophic fungi have been discussed in this review, it is also interesting to study and broaden the knowledge of other secondary metabolites that fungi are capable of exuding…”
Answer: ok
Line 573: “…described over the years, this field…”
Answer: ok
Line 575: “…secondary metabolite production…”
Answer: ok
Line 580: “…due to the absence of satisfactory matches…”
Answer: ok
Line 597: “Subsequently, following the complete chemical and biological…”
Answer: ok
Line 603-4: “Finally, due to changes in environmental conditions, mainly the result of climate change, fungal pathogens…”
Answer: ok
Line 615-16: “…biologists and chemists who have worked in the field of plant protection…”
Answer: ok
Author Response
The work under review is a review of works (152 publications!), devoted to the chemical and biological characteristics of phytotoxins produced by a number of well-known necrotrophic fungi pathogens of legumes, as well as the role of phytotoxins in the interaction of host plants and parasitic fungi.Undoubtedly, the topic is relevant both from the point of view of fundamental phytopathology and from the point of view of practice, and it will be of interest to students, university professors, and lecturers.
Answer: We thank the reviewer for these positive comments on our work
But, that's how interesting it will be for phytopathologists, this is a question!Indeed, the review is a simple compilation of published data, grouped according to the species of necrotrophic fungi, the work lacks the author's view on the essence of the issues under consideration.This could be expected in the section "General Conclusions and Perspectives", but here it only talks about the need to expand research into the study of the chemical and biological characteristics of phytotoxins and their role in the interaction of plants and parasitic fungi.
Answer: The “General Conclusions and Perspective” has been improved further by discussing the following point: 1) the use of phytotoxins in chemotaxonomy; 2) the potential application of fungal phytotoxins as a biomarker and their importance to develop biocontrol strategies; 3) the integration of metabolites data in multi-omics approach to shed lights in host-pathogen interaction mechanisms.
It should also be pointed out that there are about 40 works in the list of cited works - these are the works of the authors of the manuscript.
Answer: Sixteen new references has been added. The presence of citations of the manuscript's authors is not because of an excess of self-citation but because this review is co-authored by leading scientists in the study of the involvement of phytotoxins in the diseases of the crops under consideration, having more than 30 years of experience in the field. We cannot omit relevant citations just because are coauthored by one of the coauthors is this review. There are also about 120 other citations from other authors. We area most open to omit any citation that the reviewer might explicitly point as redundant, or to add anyone considered missing, whoever the coauthors might be.
Reviewer 4 Report
Dear Colleagues. There are some comments, they are in the file.
Author Response
We were unable to reply to reviewer 4, as no document files were added during the revision process.
Reviewer 5 Report
I recommend a revision of this review paper (as outlined in the attachment) to acknowledge previous findings in order to meet standards of writing review papers and more importantly to be more informative for readers around the world.
Ignoring recent findings on influential factors on the pathosystems reviewed by such papers could be misleading your audience, thus, it is much desired to cite valuable reports to signify our future plans/requirements for research programs

Author Response
- Comment on lines 188-189 and 205:
Authors: The sentence has been modified, and the acronyms SAR have been clarified.
- Comment on line 220:
Authors: Information on Ascochyta spp. pathosystem has been added to section 2.1. The following citation has been added:
Comparison of the epidemiology of Ascochyta blights on grain legumes. Bernard Tivoli and Sabine Banniza Eur J Plant Pathol (2007) 119:59–76 DOI 10.1007/s10658-007-9117-9.
- Comments on lines 279 -281:
Authors: The suggested references about Macrophomina- bean pathosystem have been added:
- Naseri, B. (2014). Charcoal rot of bean in diverse cropping systems and soil environments. Journal of Plant Diseases and Protection, 121(1), 20-25.
- Naseri, B. (2008). Root rot of common bean in Zanjan, Iran: major pathogens and yield loss estimates. Australasian Plant Pathology, 37(6), 546-551.
- Comment at line 291:
Authors: The following review about the sustainable control of Charcoal Rot in legumes has been added:
Naseri, B., Veisi, M., & Khaledi, N. (2018). Towards a better understanding of agronomic and soil basis for possible charcoal root rot control and production improvement in bean. Archives of Phytopathology and Plant Protection, 51(7-8), 349-358.
- Comment at line 293:
Authors: The statement about biocontrol has been modified and the citation 99 modified with the suggested review: Naseri, B., & Younesi, H. (2021). Beneficial microbes in biocontrol of root rots in bean crops: A meta-analysis (1990–2020). Physiological and Molecular Plant Pathology, 116, 101712, has now
- Comment on line 382:
Authors: Information on Colletotrichum spp. pathosystem has been added to section 2.4. The following citation has now been included: Chongo, G.; Bernier, C.C. Effects of host, inoculum concentration, wetness duration, growth stage and temperature on anthracnose of lentil. Plant Dis. 2000, 84, 544–548, doi:10.1094/PDIS.2000.84.5.544; and: Colletotrichum – current status and future directions. P.F. Cannon1*, U. Damm2 , P.R. Johnston3 , and B.S. Weir. 2012. Studies in Mycology 73: 181–213
- Comment on line 520:
Authors: We thank the reviewer for this comment; however, an extensive commentary on agronomic and environmental factors affecting legume rot epidemics is out of the aim of our work, which is focused on phytotoxins and their impact on the diseases. Nevertheless, a more comprehensive explanation of how phytotoxins could be used to develop a sustainable crop management strategy is now given in the section “4.General Conclusion and Perspectives”.
Round 2
Reviewer 3 Report
I am grateful to the authors of the manuscript for the understanding my position and for the additions they made.
Author Response
Authors: Thank you for your comments and corrections on our manuscript. We are very grateful for the constructive advices given.
Reviewer 4 Report
Dear colleagues, the term SAR (systemic acquired resistance) has been used in phytopathology and plant protection since 1961. It is not necessary to introduce confusion. Try to come up with another abbreviation for structure-activity relationship (line 222).
Author Response
Authors: : We are grateful to Reviewer 4 for this observation. Reviewer 4 is right as, unfortunately, SAR abbreviation is referred both to the structure-activity relationship between the chemical structure of a molecule and its biological activity, and also to the systemic acquired resistance as a mechanism of induced defenses.
In order to avoid confusion, we decided to delete along the text the abbreviation, and only leave the full name "structure-activity relationship".
Reviewer 5 Report
good job, the paper is ready for publication after revision based on reviewers comments to receive much more attention from experts worldwide.
Author Response
Authors: We are grateful to Reviewer 5 for the comments and suggestions to our manuscript, which have considerably improved the quality of our publication.